# Dynamic gain and frequency comb formation in exceptional-point lasers

Xingwei Gao [1] ✉, Hao He [1], Scott Sobolewski[1], Alexander Cerjan [2] ✉ & Chia Wei Hsu[1]

Exceptional points (EPs)—singularities in the parameter space of non-Hermitian systems where two nearby eigenmodes coalesce—feature unique properties with applications such as sensitivity enhancement and chiral emission. Existing realizations of EP lasers operate with static populations in the gain medium. By analyzing the full-wave Maxwell–Bloch equations, here we show that in a laser operating sufficiently close to an EP, the nonlinear gain will spontaneously induce a multi-spectral multi-modal instability above a pump threshold, which initiates an oscillating population inversion and generates a frequency comb. The efficiency of comb generation is enhanced by both the spectral degeneracy and the spatial coalescence of modes near an EP. Such an "EP comb" has a widely tunable repetition rate, self-starts without external modulators or a continuous-wave pump, and can be realized with an ultra-compact footprint. We develop an exact solution of the Maxwell–Bloch equations with an oscillating inversion, describing all spatiotemporal properties of the EP comb as a limit cycle. We numerically illustrate this phenomenon in a 5-µm-long gain-loss coupled AlGaAs cavity and adjust the EP comb repetition rate from 20 to 27 GHz. This work provides a rigorous spatiotemporal description of the rich laser behaviors that arise from the interplay between the non-Hermiticity, nonlinearity, and dynamics of a gain medium.

An exceptional point (EP) is a non-Hermitian degeneracy where not only do two eigenvalues coincide, but the spatial profiles of the two modes also become identical[1–5]. Realizing such non-Hermitian phenomena at steady-state necessitates gain and loss, making microcavity lasers a fertile ground to explore EPs. The mode coalescence and corresponding topology of the eigenvalue landscape bestow EP lasers with unique properties such as reversed pump dependence[6], loss-induced lasing[7], single-mode operation[8,9], chiral emission[10–12], sensitivity enhancement[13–18], spectral phase transitions[19], and topological state transfer[20]. In semiconductor microcavity lasers, the frequency separation between lasing modes is typically large enough that the cross beats between modes oscillate so fast that they average away before the gain medium can respond, leading to a static population inversion in the gain medium[21]. Previous realizations of EP lasers operated in this regime, yielding stable single-mode or few-mode behavior (Fig. 1a); these static-inversion lasers can be modeled by the "steady-state ab initio laser theory" (SALT)[22–26].

To enhance the performance of EP-related phenomena, such as the sensitivity of EP sensors[13–18], it is desirable to operate as close to an EP as possible. However, sufficiently close to an EP, the vanishingly small eigenvalue difference (namely, frequency difference) means that any two lasing modes of a multimode system necessarily produce beat notes slow enough to render the population inversion nonstationary. In general, since the population inversion determines the laser's gain, any nonstationary inversion produced by beat notes acts as a periodic modulation over the effective complex refractive index of the laser system. If gain's periodic modulation frequency matches the cavity's free spectral range (FSR) such that its high-quality resonances can be

[1]Ming Hsieh Department of Electrical and Computer Engineering, University of Southern California, Los Angeles, CA 90089, USA. [2]Center for Integrated Nanotechnologies, Sandia National Laboratories, Albuquerque, NM 87185, USA. ✉e-mail: xingweig@usc.edu; awcerja@sandia.gov

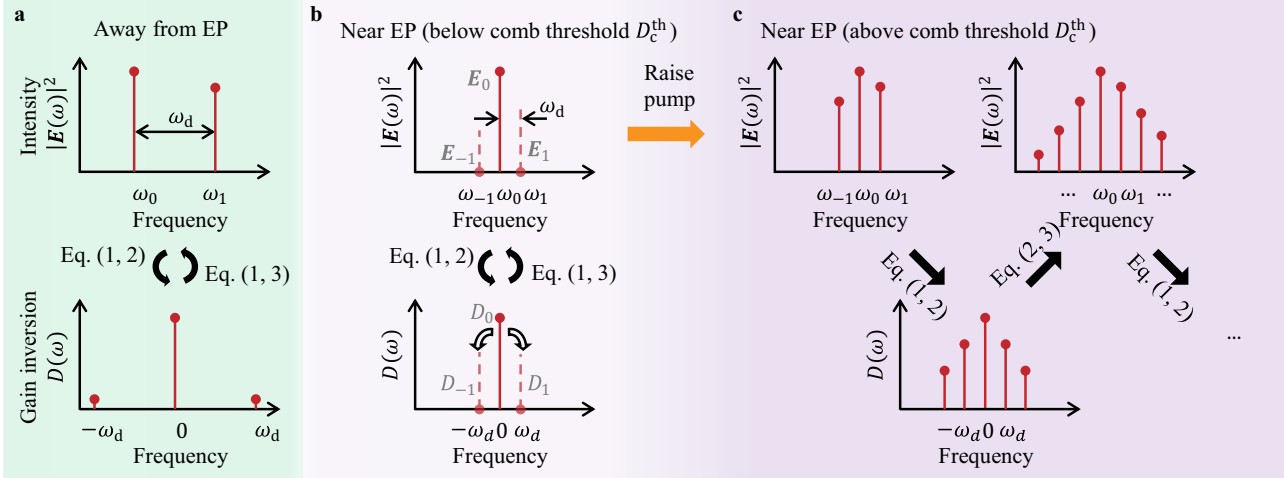

**Fig. 1 | Frequency comb formation in an exceptional-point (EP) laser. a** In an ordinary microcavity laser away from an EP, a second mode turns on when another resonance of the cavity receives enough gain to overcome its loss. Given the large frequency difference $\omega_d$, the beating between the two modes is too fast to induce significant dynamics in the population inversion $D$. **b** An EP boosts the dynamic inversion factor $\zeta \approx |\mathbf{E}_{-1}|^2/|\mathbf{E}_1|^2$ of Eq. (7), creating a multi-spectral multi-modal perturbation (dashed lines) that induces a dynamic gain oscillating at the beat frequency $\omega_d$. **c** At pumping strengths above the comb threshold $D_c^{th}$, the multi-spectral perturbation grows into sustained oscillations (solid lines), which induce additional gain oscillations and cascade down to generate a comb at frequencies $\omega_m = \omega_0 + m\omega_d$. The large-time behavior is described self-consistently by periodic-inversion ab initio laser theory (PALT) of Eqs. (11–12).

excited, a frequency comb will form whose line spacing relies on the optical size of the cavities, examples of which include mode-locked laser combs[27], Kerr combs[28,29], electro-optic combs[30], and quantum cascade laser combs[31–33]. Thus, a laser operating close to an EP has two competing frequency scales, one set by the eigenvalue splitting, and the other set by the cavity FSR; the former suggests that the system should become a comb due to population inversion dynamics, while the latter suggests that comb lines forming away from the cavity resonances will experience too much radiative loss to persist. Unfortunately, existing theories[34–44] cannot describe both the spatial complexity and the temporal dynamics of the laser in this nonstationary-inversion regime.

In this work, we develop a rigorous analysis of the full-wave Maxwell–Bloch equations and show that sufficiently close to an EP, a laser necessarily develops into a frequency comb when pumped above a comb threshold $D_c^{th}$. In this operating regime, the nonlinear gain induces a multi-spectral multi-modal perturbation that destabilizes single-mode operation and initiates temporal oscillations in the population inversion (Fig. 1b). The dynamic inversion then nonlinearly couples different frequencies to produce a frequency comb above $D_c^{th}$ (Fig. 1c). As such, our framework demonstrates that a comb must form even though the frequency of modulation driven by the dynamic inversion is typically orders of magnitude smaller than the FSR of the laser cavity. Distinct from conventional combs, such an "EP comb" has a repetition rate independent of FSR, which enables a widely tunable repetition rate and a compact cavity size. The EP comb oscillation self-starts, forming spontaneously above a pump threshold without an external modulator or an external continuous-wave laser. Moreover, we find the efficiency of comb teeth generation, characterized by a $\zeta$ factor that we introduce, to be enhanced by the spectral degeneracy and the spatial coalescence of the mode profile, conditions that are simultaneously met by operating near an EP. Finally, as an example, we provide full-wave solutions of an EP comb in an AlGaAs gain-loss coupled cavity that is merely 5-μm long, demonstrating a continuously tunable repetition rate from 20 to 27 GHz, about 400 times smaller than the free spectral range of this small cavity. Overall, the EP comb phenomena we predict provide a rich and unexpected intersection between non-Hermitian photonics, laser physics, nonlinear dynamics, and frequency combs.

## Results

### Dynamic inversion and comb formation near an exceptional point

To rigorously describe the wave physics and the spatiotemporal complexity of an EP laser, we adopt the Maxwell–Bloch (MB) equations[45,46]

$$\frac{\partial}{\partial t}D = -\gamma_{\parallel}(D - D_p) - \frac{i\gamma_{\parallel}}{2}(\mathbf{E}^* \cdot \mathbf{P} - \mathbf{E} \cdot \mathbf{P}^*), \tag{1}$$

$$\frac{\partial}{\partial t}\mathbf{P} = -(i\omega_{ba} + \gamma_{\perp})\mathbf{P} - i\gamma_{\perp}D(\mathbf{E} \cdot \boldsymbol{\theta})\boldsymbol{\theta}^*, \tag{2}$$

$$-\nabla \times \nabla \times \mathbf{E} - \frac{1}{c^2}\left(\varepsilon_c\frac{\partial^2}{\partial t^2} + \frac{\sigma}{\varepsilon_0}\frac{\partial}{\partial t}\right)\mathbf{E} = \frac{1}{c^2}\frac{\partial^2}{\partial t^2}\mathbf{P}. \tag{3}$$

The electrical field $\mathbf{E}(\mathbf{r}, t)$ is described classically with Maxwell's equations. The gain medium is described quantum mechanically as an ensemble of two-level atoms, leading to a population inversion $D(\mathbf{r}, t)$ and inducing a polarization density $\mathbf{P}(\mathbf{r}, t)$ that couple nonlinearly with $\mathbf{E}(\mathbf{r}, t)$ through dipole interactions (Supplementary Sec. 1). The $D$, $\mathbf{E}$, and $\mathbf{P}$ here are dimensionless as they have been normalized by $R^2/(\varepsilon_0\hbar\gamma_{\perp})$, $2R/(\hbar\sqrt{\gamma_{\perp}\gamma_{\parallel}})$, and $2R/(\varepsilon_0\hbar\sqrt{\gamma_{\perp}\gamma_{\parallel}})$, respectively, with $R$ being the amplitude of the atomic dipole moment, $\varepsilon_0$ the vacuum permittivity, $\hbar$ the Planck constant, and $\gamma_{\perp}$ the dephasing rate of the gain-induced polarization (i.e., the bandwidth of the gain). Here, $D_p(\mathbf{r})$ is the normalized net pumping strength and profile, $\omega_{ba}$ is the frequency gap between the two atomic levels, $\boldsymbol{\theta}$ is the unit vector of the atomic dipole moment with $\boldsymbol{\theta} \cdot \boldsymbol{\theta}^* = 1$, $\varepsilon_c(\mathbf{r})$ is the relative permittivity profile of the cold cavity, $\sigma(\mathbf{r})$ is a conductivity profile that produces linear absorption, $c$ is the vacuum speed of light. $\mathbf{E}$ and $\mathbf{P}$ satisfy an outgoing boundary condition outside the cavity.

When the pumping strength reaches the first lasing threshold $D_1^{th}$, the gain overcomes the radiation loss and absorption loss, and a single-mode lasing state $\mathbf{E}(\mathbf{r}, t) = \mathbf{E}_0(\mathbf{r})e^{-i\omega_0 t}$ emerges at a real-valued frequency $\omega_0$. Substituting this single-mode solution into the MB

equations (Supplementary Sec. 2), we get

$$\hat{O}(\omega_0)\mathbf{E}_0(\mathbf{r}) \equiv \left[-\nabla \times \nabla \times + \frac{\omega_0^2}{c^2}\varepsilon_{\text{eff}}(\mathbf{r},\omega_0)\right]\mathbf{E}_0(\mathbf{r}) = 0. \qquad (4)$$

Here, $\varepsilon_{\text{eff}}(\mathbf{r}, \omega) = \varepsilon_c(\mathbf{r}) + i\sigma(\mathbf{r})/(\omega\varepsilon_0) + \Gamma(\omega)D_0(\mathbf{r})\boldsymbol{\theta}^*\boldsymbol{\theta}\cdot$ is an effective intensity-dependent and frequency-dependent permittivity profile of the active cavity, and $\Gamma(\omega) \equiv \gamma_\perp/(\omega - \omega_{ba} + i\gamma_\perp)$. The gain $D(\mathbf{r}, t) = D_0(\mathbf{r}) = D_p(\mathbf{r})/[1 + |\Gamma(\omega_0)\mathbf{E}_0(\mathbf{r})\cdot\boldsymbol{\theta}|^2]$ is nonlinearly saturated by the local intensity, referred to as spatial hole burning. In this single-mode regime, Eq. (4) is an exact solution of the MB equations, the gain is static, and its relaxation rate $\gamma_\parallel$ plays no role at steady state.

One may freeze the nonlinearity by considering a linear operator $\hat{O}(\omega)$ in Eq. (4) that uses a fixed saturated gain $D_0(\mathbf{r}) = D_p(\mathbf{r})/[1 + |\Gamma(\omega_0)\mathbf{E}_0(\mathbf{r})\cdot\boldsymbol{\theta}|^2]$ for a fixed lasing intensity profile $|\mathbf{E}_0(\mathbf{r})|^2$. This linear $\hat{O}(\omega)$ then admits eigenmodes $\{\boldsymbol{\psi}_n(\mathbf{r})\}_n$ with complex-valued eigen-frequencies $\{\tilde{\omega}_n\}_n$, satisfying $\hat{O}(\tilde{\omega}_n)\boldsymbol{\psi}_n = 0$ with an outgoing boundary condition. We refer to them as the active-cavity resonances (also called quasinormal modes[47]). We also define operator $\hat{O}(\omega)$ below the first lasing threshold $D_1^{\text{th}}$ simply using the linear unsaturated gain $D_0(\mathbf{r}) = D_p(\mathbf{r})$. When we increase the pumping strength to $D_1^{\text{th}}$, the eigenvalue $\tilde{\omega}_0 = \omega_0$ reaches the real-frequency axis, and that resonance becomes the first lasing mode $\mathbf{E}_0(\mathbf{r}) \propto \boldsymbol{\psi}_0(\mathbf{r})$.

In the following, we define an EP as where two eigenvalues $\{\tilde{\omega}_0, \tilde{\omega}_1\}$ of the linear operator $\hat{O}(\omega)$ coalesce, at which point the corresponding mode profiles $\{\boldsymbol{\psi}_0, \boldsymbol{\psi}_1\}$ must also become the same given the non-Hermitian nature of $\hat{O}(\omega)$. An EP may exist at pumping strengths below the first lasing threshold $D_1^{\text{th}}$; such a below-threshold EP can indirectly affect laser properties[6,7,48] but cannot be directly accessed since it does not correspond to a steady-state solution. In this paper, we consider a laser close to an accessible EP at pumping strengths near or above $D_1^{\text{th}}$.

The SALT formalism assumes the population inversion to be static, $D(\mathbf{r}, t) = D_0(\mathbf{r})$[22–25]. Under SALT, the resonances $\{\boldsymbol{\psi}_n\}$ are the modes that turn on and lase when they receive enough gain. For a second mode $\boldsymbol{\psi}_1$ to turn on, it must have a spatial profile sufficiently different from the lasing mode $\mathbf{E}_0 \propto \boldsymbol{\psi}_0$ that it can amplify using the gain outside the spatial holes (i.e., away from the peaks of $|\boldsymbol{\psi}_0(\mathbf{r})|^2$). Near an EP, $\boldsymbol{\psi}_1$ necessarily has a similar spatial profile as $\boldsymbol{\psi}_0$ and so cannot turn on. Therefore, SALT predicts an EP laser to stay single-mode. However, this single-mode prediction is based on the static-inversion assumption, which is questionable near an EP since the slow beating between the two very close-by frequencies may induce dynamics in the inversion $D(\mathbf{r}, t)$. To find out what actually happens to a laser close to an EP, one must go beyond SALT and account for the inversion dynamics and its effects.

To do so, we start with a monochromatic perturbation $\mathbf{E}_1(\mathbf{r})e^{-i\omega_1 t}$ (dashed line in Fig. 1b) to single-mode operation, so the total field is $\mathbf{E}(\mathbf{r}, t) = \mathbf{E}_0(\mathbf{r})e^{-i\omega_0 t} + \mathbf{E}_1(\mathbf{r})e^{-i\omega_1 t}$. The frequency difference $\omega_d = \omega_1 - \omega_0$ can be positive or negative. With the inversion almost static, it follows from Eq. (2) that $\mathbf{P}(\mathbf{r}, t) = \mathbf{P}_0(\mathbf{r})e^{-i\omega_0 t} + \mathbf{P}_1(\mathbf{r})e^{-i\omega_1 t}$ with $\mathbf{P}_m = \Gamma_m D_0 E_m \boldsymbol{\theta}^*$ for $m = 0, 1$, where $\Gamma_m = \Gamma(\omega_m)$ and $E_m \equiv \mathbf{E}_m \cdot \boldsymbol{\theta}$. We then see from Eq. (1) that the inversion is no longer purely stationary; as illustrated in Fig. 1b, we now have $D(\mathbf{r}, t) = D_{-1}(\mathbf{r})e^{i\omega_d t} + D_0(\mathbf{r}) + D_1(\mathbf{r})e^{-i\omega_d t}$ with a dynamic component induced by the perturbation,

$$D_{-1}(\mathbf{r}) = D_1^*(\mathbf{r}) = \frac{(\Gamma_0 - \Gamma_1^*)\gamma_\parallel}{2(i\gamma_\parallel - \omega_d)}E_0(\mathbf{r})E_1^*(\mathbf{r})D_0(\mathbf{r}). \qquad (5)$$

This oscillating gain $D_{\pm 1}(\mathbf{r})e^{\mp i\omega_d t}$ arises from cross beating in the nonlinear term $\mathbf{E}^* \cdot \mathbf{P}$ of Eq. (1), so it is enhanced where $E_0(\mathbf{r})$ and $E_1(\mathbf{r})$ spatially overlap. Substituting $D(\mathbf{r}, t)$ into Eq. (2) yields a polarization $\mathbf{P}_{-1} = \Gamma_{-1}(D_{-1}E_0 + D_0 E_{-1})\boldsymbol{\theta}^*$ at new frequency $\omega_{-1} = \omega_0 - \omega_d$, which acts

like a current source to produce a new field $\mathbf{E}_{-1}$ via Eq. (3),

$$\left[-\nabla \times \nabla \times + \frac{\omega_{-1}^2}{c^2}\varepsilon_{\text{eff}}(\mathbf{r}, \omega_{-1})\right]\mathbf{E}_{-1} = -\frac{\omega_{-1}^2}{c^2}\Gamma_{-1}D_{-1}E_0\boldsymbol{\theta}^*. \qquad (6)$$

This additional frequency component $\mathbf{E}_{-1}(\mathbf{r})e^{-i\omega_{-1}t}$, generated in a four-wave-mixing[49] fashion by the nonlinear gain (Fig. 1b), differentiates an EP laser from a conventional laser and marks the onset of dynamic inversion and comb formation. To quantify the strength of this frequency generation, we solve Eq. (6) to obtain (Supplementary Sec. 3)

$$\frac{\langle|\mathbf{E}_{-1}|^2\rangle}{\langle|\mathbf{E}_1|^2\rangle} \approx \underbrace{\frac{\gamma_\parallel^2}{\omega_d^2 + \gamma_\parallel^2}\frac{\omega_0^2}{4\omega_d^2}}_{\text{I}}\underbrace{\frac{\langle|\mathbf{E}_0|^2\rangle\,|\langle D_0 E_0^3 E_1^*\rangle|^2}{\langle|\mathbf{E}_1|^2\rangle\,|\langle\varepsilon_c\mathbf{E}_0\cdot\mathbf{E}_0\rangle|^2}}_{\text{II}} \equiv \zeta, \qquad (7)$$

which we denote as the dynamic inversion factor $\zeta$. Here, $\langle \cdots \rangle = \int(\cdots)\,dr^3$ denotes integration over space. We see $\zeta$ is proportional to the lasing intensity squared, $|\mathbf{E}_0|^4$, but independent of the perturbation strength $|\mathbf{E}_1|$, so $\zeta \neq 0$ even for an infinitesimal perturbation.

The dynamic inversion factor $\zeta$ has two ingredients, Factor I on the spectral dependence, and Factor II on the spatial dependence and $|\mathbf{E}_0|^4$ laser intensity dependence. When the perturbation $\mathbf{E}_1 e^{-i\omega_1 t}$ overlaps well with the long-lived resonances $\{\boldsymbol{\psi}_n e^{-i\omega_n t}\}$, the response can sustain longer. So, the frequency difference $\omega_d = \omega_1 - \omega_0$ here correlates with the eigenvalue difference $\tilde{\omega}_1 - \tilde{\omega}_0$, which is minimized near an EP, enhancing Factor I through its $\omega_d^{-4}$ scaling. The resonances are biorthogonal with $\langle\varepsilon_c\psi_0\cdot\psi_1\rangle = 0$. As the two resonances coalesce near an EP, $\mathbf{E}_0 \propto \psi_0 \approx \psi_1$, so $\langle\varepsilon_c\mathbf{E}_0\cdot\mathbf{E}_0\rangle \approx 0$, which enhances Factor II of $\zeta$ in the same way as how an EP enhances the Petermann factor $K \equiv |\langle\varepsilon_c|\mathbf{E}_0|^2\rangle/\langle\varepsilon_c\mathbf{E}_0\cdot\mathbf{E}_0\rangle|^2$[50–57]. Such a mode coalescence promotes coupling through the stronger field overlap.

In Supplementary Sec. 4, we perform a stability analysis[58,59] to determine the decay (or growth) rate of the multi-frequency perturbation $\mathbf{E}_1(\mathbf{r})e^{-i\omega_1 t} + \mathbf{E}_{-1}(\mathbf{r})e^{-i\omega_{-1}t}$. As the pumping strength increases, the decay rate crosses over to become a growth rate, and the crossover marks the next threshold $D_2^{\text{th}}$. This is where the infinitesimal multi-frequency perturbation materializes into sustained oscillations at $\omega_{\pm 1}$. As the pump increases further, the new frequencies induce higher harmonic oscillations in the population inversion, which generates more lasing frequencies. The process cascades down to produce a frequency comb (Fig. 1c). Therefore, near an EP where the $\zeta$ factor is substantial, $D_2^{\text{th}} = D_c^{\text{th}}$ is also the threshold where the frequency comb (indicated by the subscript c) emerges, corresponding to a supercritical Hopf bifurcation[60].

We note that while the $\zeta$ factor is resonantly enhanced, the beat frequency $\omega_d$ and the coupled perturbation $\mathbf{E}_{\pm 1}(\mathbf{r})$ are determined by the linear stability eigenproblem (Supplementary Sec. 4), not by Eq. (4) as in SALT. Therefore, $\mathbf{E}_{\pm 1}(\mathbf{r})$ is generally *not* an active-cavity resonance $\psi_n$ but a superposition of multiple resonances, and the comb spacing $\omega_d$ is correlated with but not identical to the resonance spacing $|\tilde{\omega}_1 - \tilde{\omega}_0|$.

## Exact dynamic solution: PALT

The preceding analysis predicts comb formation near an EP and its threshold. To additionally predict the laser behavior above $D_c^{\text{th}}$ such as the the evolution of the comb-line intensities, repetition rate, spatial profiles, and temporal dynamics, one must address the coupling between all frequency components self-consistently. Since the cascade process couples frequencies separated by $\omega_d = \omega_1 - \omega_0$, we postulate the following spatiotemporal dependence at large time[61,62]

$$\mathbf{E}(\mathbf{r}, t) = e^{-i\omega_0 t}\sum_{m=-\infty}^{+\infty}\mathbf{E}_m(\mathbf{r})e^{-im\omega_d t}, \qquad (8)$$

$$\mathbf{P}(\mathbf{r}, t) = e^{-i\omega_0 t}\sum_{m=-\infty}^{+\infty}\boldsymbol{\theta}^* P_m(\mathbf{r})e^{-im\omega_d t}, \qquad (9)$$

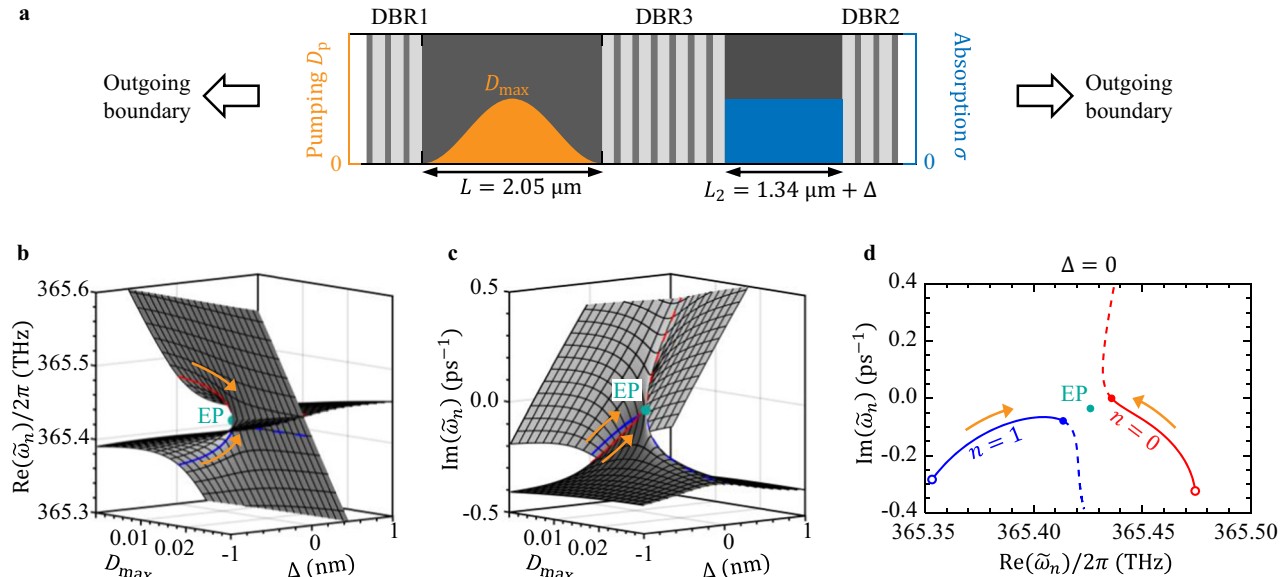

**Fig. 2 | Exceptional point in a gain-loss coupled cavity. a** A coupled 1D cavity separated by a distributed Bragg reflector (DBR), with gain in the left cavity and absorption in the right cavity. Gray-scale colors indicate the cold-cavity permittivity profile $\varepsilon_c(x)$. Orange and blue shades show the gain and absorption profiles $D_p(x)$ and $\sigma(x)$, respectively. **b, c** The two relevant eigenvalues, $\tilde{\omega}_0$ and $\tilde{\omega}_1$, of the linear operator $\hat{O}(\omega)$ in Eq. (4) with a linear gain $D_0(x) = D_p(x)$, as a function of the pumping strength $D_{max}$ and the length of the passive cavity, $L_2 = 1340$ nm + $\Delta$. The two

eigenvalues meet at an EP (green circle). The absorption in the passive cavity is $\sigma/\varepsilon_0 = 4.9$ ps$^{-1}$. The red and blue curves indicate $\tilde{\omega}_0$ and $\tilde{\omega}_1$ with $\Delta = 0$. **d** Eigenvalue trajectories on the complex-frequency plane for $\Delta = 0$, with the orange arrows indicating the directions of increasing $D_{max}$. Open circles indicate $D_{max} = 0$, and filled red and blue circles indicate the first lasing threshold $D_{max} = D_1^{th}$ where $\tilde{\omega}_0$ reaches the real-frequency axis. Dashed lines show the would-be above-threshold trajectories in the absence of gain saturation.

$$D(\mathbf{r}, t) = \sum_{m=-\infty}^{+\infty} D_m(\mathbf{r}) e^{-im\omega_d t}, \qquad (10)$$

with $\omega_0$, $\omega_d$, and $D(\mathbf{r}, t)$ being real numbers. This ansatz describes a limit cycle[60], which is periodic in time and therefore represented rigorously by a Fourier series. It also describes single-mode and two-mode operation as special cases but excludes operating with more than two cavity modes or a chaotic dynamics. The temporal periodicity is $\tau = 2\pi/\omega_d$.

We show in Sec. METHODS that the ansatz of Eqs. (8–10) forms an exact solution of the full-wave MB equations, Eqs. (1–3), with no approximation. Eliminating the gain-induced polarization yields a coupled nonlinear equation for $\{\mathbf{E}_m\}$

$$-\nabla \times \nabla \times \mathbf{E}_m + \frac{\omega_m^2}{c^2}\left(\varepsilon_c + \frac{i\sigma}{\omega_m \varepsilon_0}\right)\mathbf{E}_m = -\frac{\omega_m^2}{c^2}\Gamma_m \sum_{n=-\infty}^{+\infty} D_{m-n}(\mathbf{E}_n \cdot \boldsymbol{\theta})\boldsymbol{\theta}^*, \qquad (11)$$

and $\{D_m\}$

$$\bar{D} = D_p\left[\bar{I} - 0.5\bar{\bar{\Gamma}}_{\parallel}\left(\bar{E}^{\dagger}\bar{\bar{\Gamma}}_{+}\bar{E} - \bar{E}\bar{\bar{\Gamma}}_{-}^{\dagger}\bar{E}^{\dagger}\right)\right]^{-1}\bar{\delta}, \qquad (12)$$

with $\omega_m = \omega_0 + m\omega_d$. Different frequency components $\mathbf{E}_m$ are coherently coupled through a dynamic inversion $D_{m-n}$ oscillating at the frequency difference. Here, $\bar{D}$ and $\bar{\delta}$ are column vectors with elements $(\bar{D})_m = D_m$ and $(\bar{\delta})_m = \delta_m$, where $\delta_m$ is the Kronecker delta with $\delta_0 = 1$ and $\delta_{m\neq 0} = 0$; $\bar{I}$ is the identity matrix; $\bar{E}$ is a full matrix with elements $(\bar{E})_{mn} = \mathbf{E}_{m-n} \cdot \boldsymbol{\theta}$; $\dagger$ denotes matrix conjugate transpose; $\bar{\bar{\Gamma}}_{\parallel}$ and $\bar{\bar{\Gamma}}_{\pm}$ are diagonal matrices with $(\bar{\bar{\Gamma}}_{\parallel})_{mn} = \delta_{m-n}\gamma_{\parallel}/(m\omega_d + i\gamma_{\parallel})$ and $(\bar{\bar{\Gamma}}_{\pm})_{mn} = \delta_{m-n}\Gamma_{\pm m}$, where $\Gamma_m = \Gamma(\omega_m) = \gamma_{\perp}/(\omega_m - \omega_{ba} + i\gamma_{\perp})$ was defined earlier.

Solving Eqs. (11, 12) for $\{\mathbf{E}_m(\mathbf{r})\}$, $\{D_m(\mathbf{r})\}$, $\omega_0$, and $\omega_d$ yields all properties of the laser comb, including the frequency spectrum, temporal dynamics, spatial profiles, and input-output curves. To match the number of equations and the number of unknowns, we fix two gauge variables by recognizing that when $\mathbf{E}(\mathbf{r}, t)$ is a solution,

$e^{i\phi}\mathbf{E}(\mathbf{r}, t - t_0)$ with any real-valued $\phi$ and $t_0$ is also a solution. We name this formalism "periodic-inversion ab initio laser theory" (PALT), which overcomes the stationary-inversion limitation of SALT.

Note there is no sharp transition between an ordinary two-mode laser and an EP comb. An ordinary laser operating in the two-mode regime away from degeneracies is a trivial limit cycle with two dominant frequency components and is also rigorously described by Eqs. (8–12). Such a laser features a small $\zeta$ factor, so the second threshold $D_2^{th}$ from the stability analysis reduces to the SALT threshold (Supplementary Sec. 4), and the intensities of the additional frequency components ($m \neq 0, 1$) are small enough to be neglected. When $\zeta$ is raised, $D_2^{th}$ smoothly moves, and the additional frequency components above $D_2^{th}$ smoothly increase.

Up to now, we have considered MB equations with an ensemble of two-level atoms. In Supplementary Sec. 5, we generalize the MB equations to account for the band structure in semiconductor gain media and correspondingly generalize the PALT formalism, which does not change the conclusion on comb formation near an EP.

### EP comb example

We now use explicit full-wave examples for illustration. We adopt a parity-time-symmetric-like configuration[3,4,63], where a gain cavity is coupled to a passive cavity with material loss (Fig. 2a). Supplementary Sec. 6 lists the system parameters. The coupling and the gain-loss contrast are ingredients for an EP[1–5]. Distributed Bragg reflectors (DBRs) are used to enclose the two cavities and to separate them. The gain cavity consists of AlGaAs (refractive index $\sqrt{\varepsilon_c} = 3.4$[64], gain center $\tilde{\omega}_{ba} = 2\pi c/\omega_{ba} = 820$ nm, gain width $\gamma_{\perp} = 10^{13}$ s$^{-1}$, and relaxation rate $\gamma_{\parallel} = 10^9$ s$^{-1}$)[65]. The PALT formalism applies to any pumping profile $D_p(\mathbf{r})$; to improve the accuracy of the slow finite-difference time-domain (FDTD) simulations that we perform for validation, here we adopt a smooth profile $D_p(x) = 0.5D_{max}[1 - \cos(2\pi x/L)]$. The other cavity consists of passive GaAs ($\sqrt{\varepsilon_c} = 3.67$)[64] with a material absorption characterized by a conductivity $\sigma$. The system is homogeneous in the transverse directions ($y$ and $z$), so it reduces to a 1D problem with $\mathbf{E}_m(\mathbf{r}) = E_m(x)\hat{z}$.

In Fig. 2b, c, we show the two eigen frequencies $\{\tilde{\omega}_0, \tilde{\omega}_1\}$ of the linear operator $\hat{O}(\omega)$ of Eq. (4) as a function of the pumping strength $D_{\max}$ and the length of the passive cavity, $L_2 = 1340$ nm $+ \Delta$. To illustrate the presence of an EP, in this figure (and this figure only) we adopt a linear gain $D_0(x) = D_p(x)$ with no saturation, yielding two Riemann sheets that meet at an EP at $D_{\max} = 0.0126, \Delta = 0.01$ nm, $\tilde{\omega}_0 = \tilde{\omega}_1 = \tilde{\omega}_{EP} = 2\pi \times 365.43$ THz $- i0.0356$ ps$^{-1}$ (green circle).

Next, we fix the length of the passive cavity at $L_2 = 1340$ nm ($\Delta = 0$), for which the pump dependence of the two eigenvalues is shown by the red ($n = 0$) and blue ($n = 1$) curves in Fig. 2b–d. At pumping strength $D_{\max} = D_1^{th} = 0.0124$ (red and blue filled circles in Fig. 2d), $\tilde{\omega}_0$ reaches the real-frequency axis, and $E_0(x) \propto \psi_0(x)$ turns on as the first lasing mode; the Petermann factor there is $K_0 \equiv |\langle \varepsilon_c | \psi_0 \rangle|^2 / |\langle \varepsilon_c \psi_0^2 \rangle|^2 = 28$.

Above the first threshold ($D_1^{th} < D_{\max} < D_2^{th}$), the red and blue dashed lines in Fig. 2b–d show the would-be eigenvalue trajectories with a hypothetical linear gain, in which case the system enters a PT-broken phase where one mode is localized in the pumped cavity, and the other mode is localized in the lossy cavity. Gain saturation, however, clamps the saturated gain at the same level as the overall loss, which fixes the two nonlinearity-frozen eigenvalues $\{\omega_0, \omega_1\}$ near where they are at $D_1^{th}$ (red and blue filled circles in Fig. 2d), and this single-mode laser stays close to a nonlinear EP without entering the PT-broken phase.

As the pumping strength reaches above $D_{\max} > D_2^{th} = D_c^{th} = 0.064$, the population inversion starts to oscillate (Fig. 3e), and a frequency comb emerges (Fig. 3g). Given the proximity to an EP, the repetition rate $|\omega_d| \approx 1.35 \times 10^{11}$ rad/s at $D_c^{th}$ is around 400 times smaller than the FSR of the overall cavity, and the dynamic inversion factor $\zeta \approx 0.26$ is sizeable. For a complete characterization, we show in Fig. 4a the evolution of the intensity at different frequencies as a function of the pumping strength. To keep the frequency difference $|\omega_d|$ small, we raise the absorption level $\sigma$ when $D_{\max} > D_c^{th}$ (Fig. 4b, c). The two center comb lines $\{\omega_0, \omega_1\}$ lie close to the two near-degenerate active-cavity resonances from SALT in Eq. (4), $\{\mathrm{Re}(\tilde{\omega}_0), \mathrm{Re}(\tilde{\omega}_1)\}$; the remaining comb lines are generated by the nonlinear gain through four-wave mixing and are not lined up with any additional cavity modes (Supplementary Fig. 3). The spatial profiles at different frequencies are almost identical (Fig. 3c); they remain comparable to the profiles near the EP in the single-mode regime $D_1^{th} < D_{\max} < D_2^{th}$ where the two modes almost coalesce, without entering the PT-broken phase. Supplementary Sec. 6 shows the intensity and gain profiles at all frequencies and their relative phases.

It is commonly assumed[21,25,26,58,59] that the stationary-inversion approximation (SIA) of SALT is valid when $|\omega_d| > \gamma_\parallel$, namely when the beat notes oscillate faster than the gain relaxation rate. However, such a reasoning does not account for the EP-enhanced frequency generation, as captured by the dynamic inversion factor $\zeta$ in Eq. (7). In the present example, $|\omega_d| \approx 1.35 \times 10^{11}$ rad/s is two orders of magnitude greater than $\gamma_\parallel = 10^9$ s$^{-1}$, but SALT (blue circle in Fig. 3g and red dashed line in Fig. 4a) already fails above the comb threshold. As described in Sec. Dynamic inversion and comb formation near an exceptional point and shown in Fig. 2d, SALT predicts the laser to stay single-mode because $\psi_1(x)$ has almost the same spatial profile as the lasing mode $E_0(x) \propto \psi_0(x)$ near an EP, so it experiences the same gain clamping as $E_0(x)$ and cannot turn on; this would indeed be the laser behavior when the system is near an EP but not close enough. In the present example, given the very close proximity to an EP and the resulting large dynamic inversion factor $\zeta \approx 0.26$, what actually turns on at the comb threshold $D_2^{th} = D_c^{th}$ is not an isolated resonance $\psi_1$ of the operator $\hat{O}(\omega)$ in Eq. (4) but the multi-spectral multi-modal perturbation $E_1(x)e^{-i\omega_1 t} + E_{-1}(x)e^{-i\omega_{-1} t}$ described in Sec. Dynamic inversion and comb formation near an exceptional point, which is a superposition of multiple resonances and can amplify by additionally utilizing the dynamic gain $D_{\pm 1}(x)e^{\mp i\omega_d t}$ of Eq. (5).

As a comparison to the near-EP laser above, we also consider an ordinary single-cavity laser (Fig. 3b) sandwiched between two DBR

partial mirrors, operating in the two-mode regime. The active cavity has the same AlGaAs gain. At pumping strength $D_{\max} > D_2^{th} = 0.033$, two modes that differ by one longitudinal order lase (Fig. 3d) and produce a sinusoidal beating pattern (Fig. 3f). Here, the population inversion is static (Fig. 3f), and only two peaks appear in the spectrum (Fig. 3h). There is no EP nearby in the parameter space. The frequency separation $|\omega_d| \approx \pi c / \sqrt{\varepsilon_c} L \approx 5.4 \times 10^{13}$ rad/s equals the free spectral range (FSR) of the cavity and is over four orders of magnitude greater than $\gamma_\parallel$, leading to a negligible dynamic inversion factor $\zeta \approx 3 \times 10^{-13}$. The Petermann factor is $K = 1.0$ here; the gain only balances the radiation loss and does not introduce mode non-orthogonality. For such a two-mode laser away from degeneracies, PALT reduces to SALT (blue circles in Fig. 3h).

EPs feature a boosted sensitivity[13–18], which also amplifies the numerical error, requiring an unusually high precision when solving Eqs. (11–12). We find a finite-difference discretization[25] and the threshold constant-flux basis[24] to both require an impractically large basis to reach a satisfactory accuracy near an EP. To improve the numerical efficiency, here we develop a volume-integral formalism that employs accurate semi-analytic Green's function of the passive system to solve Eqs. (11, 12) (Supplementary Sec. 7).

To validate our prediction and to verify the stability of the single-mode and the comb solutions, we additionally carry out direct integration of the MB equations, Eqs. (1–3), using FDTD, where we evolve the system until all transient behaviors settle away (Supplementary Sec. 8). The time-consuming FDTD simulations agree quantitatively with all of the PALT predictions (Figs. 3–4). Figure 4d shows the field evolution in FDTD when the pump is raised across the comb threshold.

Since the EP comb repetition rate $f_d = |\omega_d|/(2\pi)$ is not tied to the cavity FSR, we can adjust it freely, for example, by tuning the material absorption as shown in Fig. 5. This is not possible with mode-locked combs, Kerr combs, and quantum cascade laser combs.

In the preceding example, we bring the laser close to an EP. Supplementary Sec. 9 shows that the behavior is the same when we tune the system parameters with a higher precision such that the system has an almost exact EP above the first threshold, $D_{EP} > D_1^{th}$. With increasing pump (while fixing the other system parameters), such a laser reaches the comb threshold $D_c^{th}$ and develops into a stable EP comb soon after $D_1^{th}$. The exact-EP single-mode lasing state is unreachable as it lies at a higher pump (namely, $D_{EP} > D_c^{th} \gtrsim D_1^{th}$) and is unstable.

## Discussion

In this work, we answer the question of what happens to a laser close to an EP. Based on the full-wave MB equations, we show that the spectral degeneracy and the spatial coalescence of modes near an EP work with the nonlinearity of the gain medium to induce oscillations in the population inversion, resulting in an "EP comb." The EP comb features a continuously tunable repetition rate, an ultra-compact cavity size, and a self-starting operation with no need for an external modulator or continuous-wave laser. The PALT formalism fully describes both the spatial complexity and the temporal dynamics of such a limit-cycle laser state, overcoming the stationary-inversion limitation of SALT. This EP comb phenomenon uniquely bridges the subjects of non-Hermitian photonics, laser physics, nonlinear dynamics, and frequency combs.

As EP sensors are more sensitive closer to an EP[13–18], it may be desirable to operate such a sensor as close to an EP as possible. This work shows that when an EP laser is brought sufficiently close to an EP, it necessarily develops into a comb above a pump threshold. In such a comb regime, the optimal sensing scheme and the parametric dependence are nontrivial and can be the subject of a future study.

Existing realizations of EP lasers had mode spacing above 100 GHz; given the $\omega_d^{-4}$ scaling of the dynamic inversion factor $\zeta$ in Eq. (7), the $\zeta$s there were too small to induce the multi-spectral multi-modal instability responsible for comb formation, so those lasers exhibited static single-mode behavior. Further reduction of the mode spacing requires finer tuning but is possible. In fact, the self-pulsation

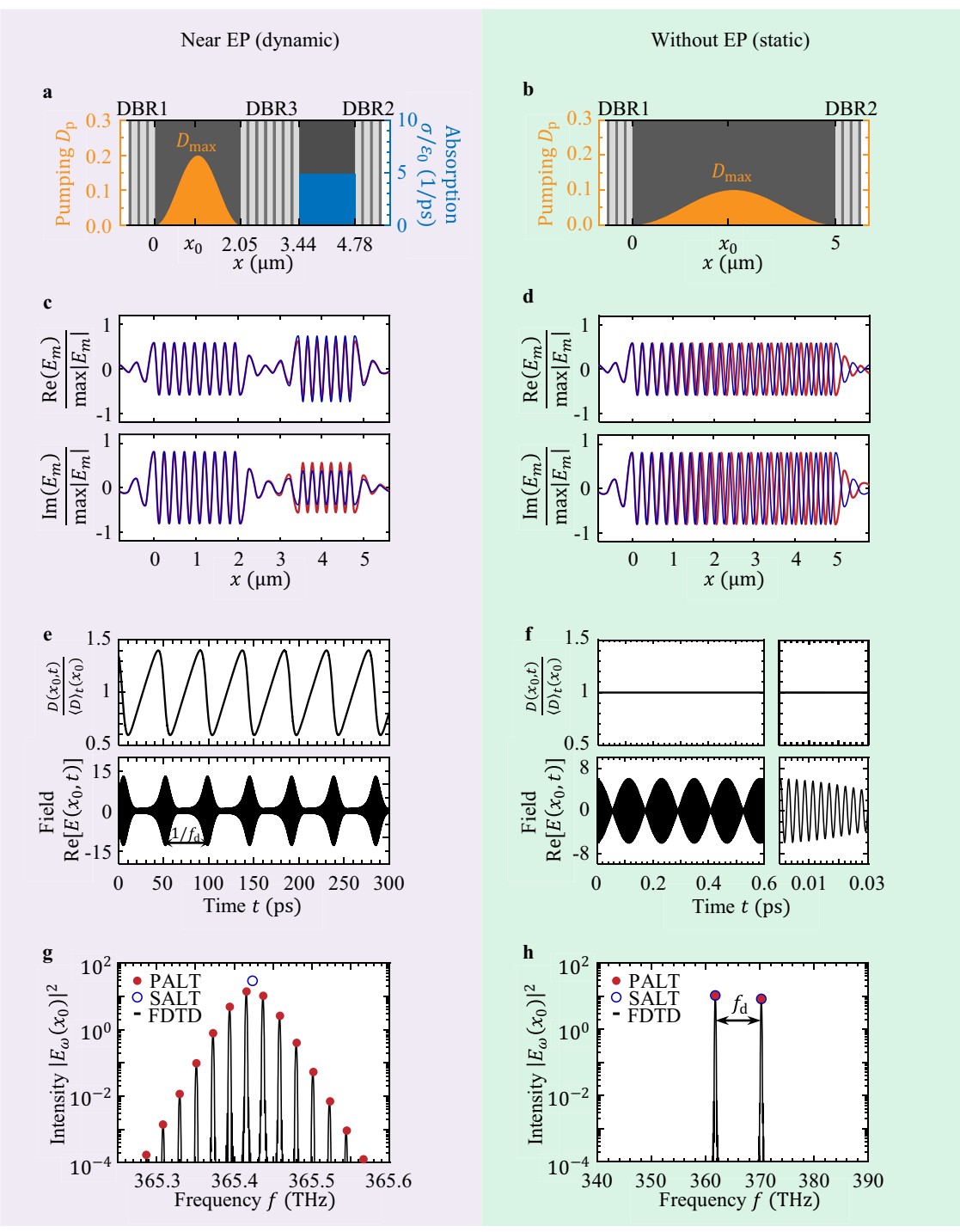

**Fig. 3 | Laser behavior above the second threshold $D_2^{th}$ near and away from an EP. a** The EP laser in Fig. 2a with $\Delta = 0$, $D_{max} = 0.2$, $\sigma/\varepsilon_0 = 8.1\ ps^{-1}$. **b** An ordinary AlGaAs laser cavity with DBR partial mirrors. **c, d** Spatial profiles $E_0$ (red) and $E_1$ (blue) at $\omega_0$ and $\omega_1$, from the full-wave PALT solution. The two profiles are orthogonal in the ordinary laser but almost identical in the EP laser. **e, f** Dynamics of the population inversion $D(x_0, t)$ and electrical field $E(x_0, t)$ at the location $x_0$ shown in (**a, b**). $\langle D \rangle_t$ denotes the inversion averaged over time. **g, h** The intensity spectrum, comparing the PALT solution to the existing "steady-state ab initio laser theory" (SALT) and to FDTD simulations of the Maxwell–Bloch equations.

observed in an InAs-quantum-dot Fano laser[66] may have been an EP comb since that system has the features of an EP comb (self-starting comb formation in a compact microcavity) and all the EP ingredients: two modes with similar frequencies (one from a line-defect waveguide and one from a nanocavity), near-field coupling between the two modes, and differential gain (as only the waveguide is pumped).

The repetition rate $|\omega_d|$ of the EP comb is determined by the stability eigenvalue problem (Supplementary Sec. 4) at the threshold $D_c^{th}$ and by solving the nonlinear Eqs. (11, 12) self-consistently above $D_c^{th}$. While it is hard to extract insights from these complex equations, empirically we found the distance between the two linear SALT eigenvalues of Eq. (4) to provide a crude approximation,

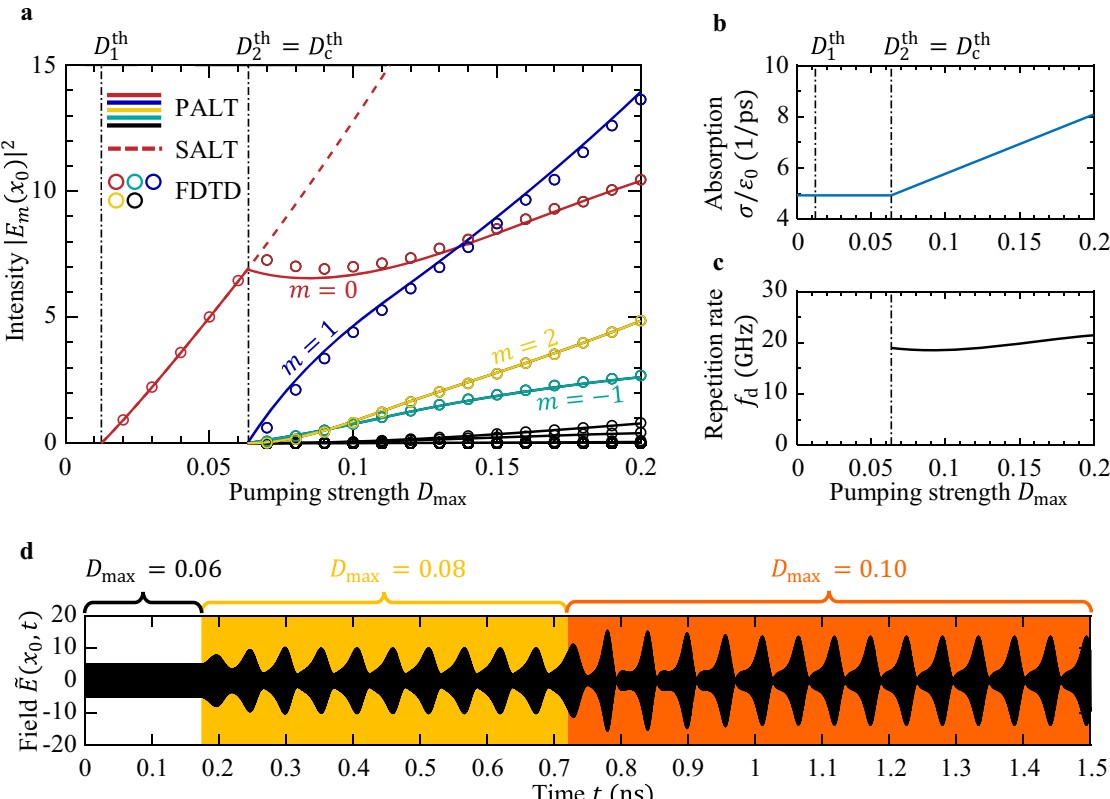

**Fig. 4 | Pump dependence of an EP laser. a** Input-output curves: lasing intensity at frequencies $f_m = f_0 + mf_d$ as a function of the pumping strength, comparing the PALT solution (solid lines) to SALT (red dashed line) and FDTD simulations (circles). **b**, **c** The level of absorption $\sigma$ is raised with the pump to keep the frequency difference $f_d$ small above the comb threshold $D_c^{th} = 0.064$. **d** Field evolution in FDTD when the pump is raised across the comb threshold.

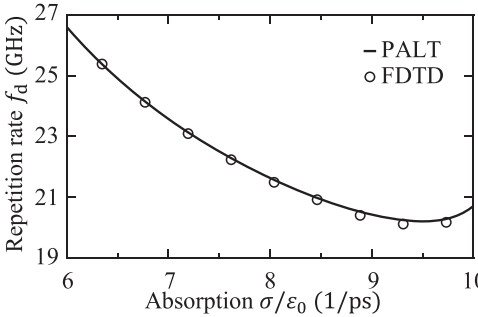

**Fig. 5 | Tunability of the EP comb.** Repetition rate of the preceding EP comb when the level of absorption is tuned while the pumping strength is fixed at $D_{max} = 0.2$.

$|\omega_d| \approx |\tilde{\omega}_1 - \tilde{\omega}_0|$, near $D_c^{th}$. Close to an EP, the two linear eigenvalues are sensitive to all parameters of the system, so $\omega_d$ can be tuned by changing the absorption, coupling strength, refractive index, etc. The repetition rate can also be reduced by considering larger cavities. The minimal repetition rate is limited by the laser linewidth, which can be reduced with standard methods.

The limit of the linewidth itself would be an interesting subject for future investigations. The divergent Petermann factor is known to broaden the linewidth[50,51,56,57,67]. With an EP comb, we expect even richer noise properties since the dynamic population inversion can modify the spontaneous emission beyond noise models that assume a linear gain[54,68] or a stationary inversion[42,51,57,69,70]. Additionally, the relation between noise and atomic populations is commonly derived

at a local thermal equilibrium[51,69], but such an equilibrium is no longer reached when the inversion fluctuates faster than the spontaneous emission rate.

An EP comb provides a doorway to other nonlinear dynamics phenomena such as bistability, period doubling, and chaos. Future work can study the stability of the EP comb, its bifurcation properties, and the transition to other dynamic regimes. The comb spectrum may be further analyzed and optimized. The PALT formalism can also describe lasers near Hermitian degeneracies due to symmetry, going beyond perturbation theory[26] and stability analysis[58,59]. We expect even richer behaviors near higher-order EPs and in spatially complex systems such as random lasers and chaotic-cavity lasers.

## Methods
### Derivation of PALT
We show that the PALT ansatz in Eqs. (8–10) forms an exact solution of the MB equations. In doing so, we also derive Eqs. (11, 12).

We substitute Eqs. (8–10) into the MB equations, Eqs. (1–3), and match terms with the same time dependence. Solving Eq. (1), we get

$$D_m = D_p \delta_m + \frac{1}{2m\omega_d + i\gamma_\parallel} \frac{\gamma_\parallel}{} \sum_{n=-\infty}^{+\infty} \left( \mathbf{E}_{-m+n}^* \cdot \boldsymbol{\theta}^* P_n - \mathbf{E}_{m-n} \cdot \boldsymbol{\theta} P_{-n}^* \right), \quad (13)$$

where $\delta_m$ is the Kronecker delta with $\delta_0 = 1$ and $\delta_{m\neq 0} = 0$. From Eq. (2), we get

$$P_m = \Gamma_m \sum_{n=-\infty}^{+\infty} D_n \mathbf{E}_{m-n} \cdot \boldsymbol{\theta}, \quad (14)$$

where $\Gamma_m = \Gamma(\omega_m) = \gamma_\perp / (\omega_m - \omega_{ba} + i\gamma_\perp)$ and $\omega_m = m\omega_{\rm d} + \omega_0$. From Eq. (3), we get

$$-\nabla \times \nabla \times \mathbf{E}_m + \frac{\omega_m^2}{c^2}\left(\varepsilon_c + \frac{i\sigma}{\omega_m \varepsilon_0}\right)\mathbf{E}_m = -\frac{\omega_m^2}{c^2}P_m \hat{\boldsymbol{\theta}}^*. \quad (15)$$

This confirms that all of the MB equations, Eqs. (1–3), are satisfied with no approximation. Substituting Eq. (14) into Eq. (15), we get Eq. (11), where we have applied the commutativity of convolution,

$$\sum_{n=-\infty}^{+\infty} D_n \mathbf{E}_{m-n} \cdot \boldsymbol{\theta} = \sum_{n=-\infty}^{+\infty} D_{m-n} \mathbf{E}_n \cdot \boldsymbol{\theta}. \quad (16)$$

To eliminate the gain-induced polarization, we first recognize that since $D(\mathbf{r}, t)$ is real-valued, its Fourier components have to be symmetric, $D_n^* = D_{-n}$. With this fact, we take the complex conjugate of Eq. (14) and then replace the dummy variable $n$ by $-n$,

$$\begin{aligned}
P_{-m}^* &= \Gamma_{-m}^* \sum_{n=-\infty}^{+\infty} D_n^* \mathbf{E}_{-m-n}^* \cdot \boldsymbol{\theta} \\
&= \Gamma_{-m}^* \sum_{n=-\infty}^{+\infty} D_{-n} \mathbf{E}_{-m-n}^* \cdot \boldsymbol{\theta} \\
&= \Gamma_{-m}^* \sum_{n=-\infty}^{+\infty} D_n \mathbf{E}_{-m+n}^* \cdot \boldsymbol{\theta}.
\end{aligned} \quad (17)$$

Eqs. (13, 14, 17) can be summarized in matrix form as

$$\bar{D} = D_{\rm p}\bar{\delta} + \frac{1}{2}\bar{\bar{\Gamma}}_\parallel (\bar{\bar{E}}^\dagger \bar{P} - \bar{\bar{E}}\bar{P}_-^*), \quad (18)$$

$$\bar{P} = \bar{\bar{\Gamma}}_+ \bar{\bar{E}}\bar{D}, \quad (19)$$

$$\bar{P}_-^* = \bar{\bar{\Gamma}}_- \bar{\bar{E}}^\dagger \bar{D}, \quad (20)$$

where † denotes matrix conjugate transpose, with
- Column vectors: $(\bar{P})_m = P_m$, $(\bar{P}_-^*)_m = P_{-m}^*$, $(\bar{D})_m = D_m$, and $(\bar{\delta})_m = \delta_m$.
- Matrices: $(\bar{\bar{E}})_{mn} = \mathbf{E}_{m-n} \cdot \boldsymbol{\theta}$, $(\bar{\bar{\Gamma}}_\parallel)_{mn} = \delta_{m-n}\gamma_\parallel / (m\omega_{\rm d} + i\gamma_\parallel)$, $(\bar{\bar{\Gamma}}_\pm)_{mn} = \delta_{m-n}\Gamma_{\pm m}$.

Substituting Eqs. (19, 20) into Eq. (18), we can solve for $\bar{D}$ to obtain Eq. (12).

## Slow-gain limit

In the slow-gain limit of $|\omega_d| \gg \gamma_\parallel$, all entries of the diagonal matrix $\bar{\bar{\Gamma}}_\parallel$ are approximately 0 except $(\bar{\bar{\Gamma}}_\parallel)_{00} = -i$. In this limit, Eq. (12) simplifies to $\bar{D} \approx D_0 \bar{\delta}$ with

$$D_0(\mathbf{r}) \approx \frac{D_{\rm p}(\mathbf{r})}{1 + \sum_m |\Gamma_m \mathbf{E}_m(\mathbf{r}) \cdot \boldsymbol{\theta}|^2}. \quad (21)$$

Then, Eq. (19) yields $P_m \approx \Gamma_m D_0 (\mathbf{E}_m \cdot \boldsymbol{\theta})$, so Eq. (11) becomes

$$\left[-\nabla \times \nabla \times + \frac{\omega_m^2}{c^2}\left(\varepsilon_c + \frac{i\sigma}{\omega_m \varepsilon_0} + \Gamma_m D_0 \boldsymbol{\theta}^* \boldsymbol{\theta}\cdot\right)\right]\mathbf{E}_m \approx 0. \quad (22)$$

These Eqs. (21, 22) reduce to SALT[24,25] in the single-mode or two-mode regime (with two indices, $m = 0, 1$; $\mathbf{E}_m$ with $m \neq 0$ or 1 has to be zero unless $\omega_m$ happens to be the resonant frequency of a third lasing mode).

## Fast-gain limit

In the fast-gain limit where the lasing bandwidth is much smaller than both $\gamma_\parallel$ and $\gamma_\perp$, we can show that $\bar{\bar{\Gamma}}_\parallel \approx -i\bar{\bar{I}}$, $\bar{\bar{\Gamma}}_\pm \approx \Gamma_0 \bar{\bar{I}}$, and $\bar{\bar{E}}^\dagger \bar{\bar{E}} = \bar{\bar{E}}\bar{\bar{E}}^\dagger$. In this limit, Eq. (12) simplifies to

$$\bar{D} \approx D_{\rm p}(\bar{\bar{I}} + |\Gamma_0|^2 \bar{\bar{E}}^\dagger \bar{\bar{E}})^{-1}\bar{\delta}, \quad (23)$$

where we have applied $\Gamma_0 - \Gamma_0^* = -2i|\Gamma_0|^2$. Eq. (23) yields

$$\bar{D} + |\Gamma_0|^2 \bar{\bar{E}}^\dagger \bar{\bar{E}}\bar{D} \approx D_{\rm p}\bar{\delta}. \quad (24)$$

The entries of the column vector $\bar{\bar{E}}^\dagger \bar{\bar{E}}\bar{D}$ are the Fourier components of $|\mathbf{E}(\mathbf{r},t) \cdot \hat{\boldsymbol{\theta}}|^2 D(\mathbf{r},t)$. Therefore, if we multiply Eq. (24) to the left with the row vector $[..., e^{2i\omega_{\rm d}t}, e^{i\omega_{\rm d}t}, 1, e^{-i\omega_{\rm d}t}, e^{-2i\omega_{\rm d}t}, ...]$, we obtain the time evolution $D(\mathbf{r}, t) + |\Gamma_0 \mathbf{E}(\mathbf{r}, t) \cdot \hat{\boldsymbol{\theta}}|^2 D(\mathbf{r}, t) \approx D_{\rm p}(\mathbf{r})$, namely

$$D(\mathbf{r}, t) \approx \frac{D_{\rm p}(\mathbf{r})}{1 + |\Gamma_0 \mathbf{E}(\mathbf{r}, t) \cdot \hat{\boldsymbol{\theta}}|^2}. \quad (25)$$

In this fast-gain limit, the instantaneous population inversion is given by the instantaneous intensity at that time.

## Data availability

The data of PALT calculation and FDTD simulation results presented in the paper are available on OSF database [https://osf.io/jptza/].

## Code availability

Codes that reproduce the results in this study, including the PALT integral equation solver, stability eigenvalue solver, and Maxwell–Bloch FDTD simulations, are available on GitHub [https://github.com/complexphoton/PALT].

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

## Acknowledgements

We thank M. Khajavikhan, A. D. Stone, L. Ge, S. G. Johnson, and M. Yu for helpful discussions. This work was supported by the National Science Foundation CAREER award (ECCS-2146021) and the University of Southern California. A.C. acknowledges support from the US Department of Energy, Office of Basic Energy Sciences, Division of Materials Sciences and Engineering. This work was performed, in part, at the Center for Integrated Nanotechnologies, an Office of Science User Facility operated by the US Department of Energy (DOE) Office of Science. Sandia National Laboratories is a multimission laboratory managed and operated by National Technology & Engineering Solutions of Sandia, LLC, a wholly-owned subsidiary of Honeywell International, Inc., for the US DOE's National Nuclear Security Administration under contract DE-NA-0003525. The views expressed in the article do not necessarily represent the views of the US DOE or the United States Government.

## Author contributions

X.G. developed the theory on the ζ factor, stability analysis, PALT, integral equation solver, and performed the SALT and PALT calculations. H.H., assisted by A.C., developed the FDTD code. X.G., H.H., and S.S. performed the FDTD simulations. H.H. developed the spectral analysis of the FDTD results. C.W.H. conceived of the project. C.W.H. and A.C. supervised the project. C.W.H. and X.G. wrote the paper with inputs from the other coauthors. All authors discussed the results.

## Competing interests

The authors declare no competing interests.
