## [Peer Review File · Nature Communications]

REVIEWER COMMENTS

Reviewer #1 (Remarks to the Author):

The authors have made significant improvements to the manuscript, addressing all of my concerns. While the experimental demonstration remained a point of contention, the authors conveniently addressed this by identifying a relevant paper on Fano-lasers published in *Nature Photonics*. Having familiarity with this work, I concur with the authors who establishes a connection between their spontaneous frequency combs and the Fano-laser. In this sense, I recommend incorporating the three "ingredients" outlined by the authors in their response into the manuscript. This addition will enrich the manuscript relevance.

Furthermore, I suggest relocating the new supplementary Fig. 3 to the main text. This figure not only elucidates the parameters utilized to tune the laser but also serves as a highly informative visual aid, particularly in a paper focusing on exceptional points (EP). By integrating this figure into the main body of the manuscript, the clarity of the study will be significantly enhanced.

In Summary, in my opinion the manuscript can be accepted for publication in *Nature Communications* with the above optional recommendations.

Reviewer #2 (Remarks to the Author):

In the manuscript, the authors introduce the periodic-inversion ab initio laser theory (PALT), which describes the behavior of multi-mode lasing near an exceptional point (EP), characterized by the coalescence of two eigenmodes. Same as in other non-Hermitian systems, the EP in a laser cavity emerges from the gain-loss tuning of the media. And this process is described by Eq. 4, as the population inversion density resides between the first and second lasing thresholds ($D_1 < D < D_2$).

This manuscript is well-motivated, self-consistent, and of high quality, making me pleased to recommend it for publication. However, before making a final decision, I have one question and one suggestion to offer.

Suggestion:

I suggest adding a subfigure to Fig. 1, positioned before the existing two subfigures. The content for it can refer to Fig. S3c and the draft figure I have submitted. The purpose is to visually emphasize the key

sentence in the first page:

"We identify two ingredients that promote comb formation: spectral degeneracy and the spatial coalescence of the mode profile."

With this subfigure, we can clearly depict spectral degeneracy (or close to it) through labeling small ω_d and suggest spatial coalescence through the intersecting eigen-spectra that characterize EP.

Question:

EP is also recognized as a phase transition point, such as the transition between PT-exact/broken phase. Even without PT symmetry, the discontinuous eigen-spectra (Fig. S3c) signify a transition in mode profiles. Is the frequency comb influenced by this transition, or just by ω_d ?

Reviewer #3 (Remarks to the Author):

In this revised manuscript, the authors have effectively addressed the majority of the issues I raised. However, before recommending this paper for publication, I still have two concerns based on the authors' response.

Firstly, with regard to the potential impact of the research, the authors assert that the EP laser can be utilized for sensing. While it is acknowledged that enhanced sensitivity has been demonstrated using microlasers at EPs (Refs. [15,16]), there appears to be some ambiguity regarding the specific sensing improvements facilitated by the current work.

Secondly, the authors have clarified that the repetition rate of the EP comb can be adjusted by varying different parameters. Nevertheless, the underlying physics governing this dependence remains unclear. It raises questions as to whether the comb lines are not in resonance with the cavity modes. Further clarification on this aspect would be beneficial for a comprehensive understanding of the findings.

Response to Reviewer #1

The authors have made significant improvements to the manuscript, addressing all of my concerns.

Response: We thank the reviewer for the valuable feedback, which helped us improve the manuscript.

While the experimental demonstration remained a point of contention, the authors conveniently addressed this by identifying a relevant paper on Fano-lasers published in Nature Photonics. Having familiarity with this work, I concur with the authors who establishes a connection between their spontaneous frequency combs and the Fano-laser. In this sense, I recommend incorporating the three "ingredients" outlined by the authors in their response into the manuscript. This addition will enrich the manuscript relevance.

Response: We thank the reviewer for this suggestion. We now expanded the following sentence in the Discussion section (page 8 of the main text) to incorporate these three ingredients:

In fact, the self-pulsation observed in an InAs-quantum-dot Fano laser⁶⁴ may have been an EP comb since that system has the features of an EP comb (self-starting comb formation in a compact microcavity) and all the EP ingredients: two modes with similar frequencies (one from a line-defect waveguide and one from a nanocavity), near-field coupling between the two modes, and differential gain (as only the waveguide is pumped).

Furthermore, I suggest relocating the new supplementary Fig. 3 to the main text. This figure not only elucidates the parameters utilized to tune the laser but also serves as a highly informative visual aid, particularly in a paper focusing on exceptional points (EP). By integrating this figure into the main body of the manuscript, the clarity of the study will be significantly enhanced.

Response: We thank the reviewer for this suggestion. We now relocate Supplementary Fig. 3 to the main text, as Fig 2 on page 5:

Fig. 2. Exceptional point in a gain-loss coupled cavity. **a**, A coupled 1D cavity separated by a distributed Bragg reflector (DBR), with gain in the left cavity and absorption in the right cavity. Gray-scale colors indicate the cold-cavity permittivity profile $\epsilon_c(x)$. Orange and blue shades show the gain and absorption profiles $D_p(x)$ and $\sigma(x)$, respectively. **b,c**, The two relevant eigenvalues, $\tilde{\omega}_0$ and $\tilde{\omega}_1$, of the linear operator $\hat{O}(\omega)$ in Eq. (4) with a linear gain $D_0(x) = D_p(x)$, as a function of the pumping strength D_{\max} and the length of the passive cavity, $L_2 = 1340 \text{ nm} + \Delta$. The two eigenvalues meet at an EP (green circle). The absorption in the passive cavity is $\sigma/\epsilon_0 = 4.9 \text{ ps}^{-1}$. The red and blue curves indicate $\tilde{\omega}_0$ and $\tilde{\omega}_1$ with $\Delta = 0$. **d**, Eigenvalue trajectories on the complex-frequency plane for $\Delta = 0$, with the orange arrows indicating the directions of increasing D_{\max} . Open circles indicate $D_{\max} = 0$, and filled red and blue circles indicate the first lasing threshold $D_{\max} = D_1^{\text{th}}$ where $\tilde{\omega}_0$ reaches the real-frequency axis. Dashed lines show the would-be above-threshold trajectories in the absence of gain saturation.

Correspondingly, we also brought the formal definition of these eigenvalues to the main text by defining operator $\hat{O}(\omega)$ in Eq. (4) on page 2:

$$\hat{O}(\omega_0) \mathbf{E}_0(\mathbf{r}) \equiv \left[-\nabla \times \nabla \times + \frac{\omega_0^2}{c^2} \epsilon_{\text{eff}}(\mathbf{r}, \omega_0) \right] \mathbf{E}_0(\mathbf{r}) = 0.$$

and expanding the following sentences on page 3:

One may “freeze” the nonlinearity by considering a linear operator $\hat{O}(\omega)$ in Eq. (4) that uses a fixed saturated gain $D_0(\mathbf{r}) = D_p(\mathbf{r})/[1+|\Gamma(\omega_0) \mathbf{E}_0(\mathbf{r}) \cdot \boldsymbol{\theta}|^2]$ for a fixed lasing intensity profile $|\mathbf{E}_0(\mathbf{r})|^2$. This linear $\hat{O}(\omega)$ then admits eigenmodes $\{\boldsymbol{\psi}_n(\mathbf{r})\}_n$ with complex-valued eigenfrequencies $\{\tilde{\omega}_n\}_n$, satisfying $\hat{O}(\tilde{\omega}_n) \boldsymbol{\psi}_n = 0$ with an outgoing boundary condition. We refer to them as the active-cavity resonances (also called quasinormal modes⁴⁷). We also define operator $\hat{O}(\omega)$ below the first lasing threshold D_1^{th} simply using the linear unsaturated gain $D_0(\mathbf{r}) = D_p(\mathbf{r})$.

The following two paragraphs on page 5 of the main text describe the added Fig. 2:

In Fig. 2b–c, we show the two eigenfrequencies $\{\tilde{\omega}_0, \tilde{\omega}_1\}$ of the linear operator $\hat{O}(\omega)$ of Eq. (4) as a function of the pumping strength D_{\max} and the length of the passive

cavity, $L_2 = 1340 \text{ nm} + \Delta$. To illustrate the presence of an EP, in this figure (and this figure only) we adopt a linear gain $D_0(x) = D_p(x)$ with no saturation, yielding two Riemann sheets that meet at an EP at $D_{\text{max}} = 0.0126$, $\Delta = 0.01 \text{ nm}$, $\tilde{\omega}_0 = \tilde{\omega}_1 = \tilde{\omega}_{\text{EP}} = 2\pi \times 365.43 \text{ THz} - i0.0356 \text{ ps}^{-1}$ (green circle).

Next, we fix the length of the passive cavity at $L_2 = 1340 \text{ nm}$ ($\Delta = 0$), for which the pump dependence of the two eigenvalues is shown by the red ($n = 0$) and blue ($n = 1$) curves in In Fig. 2b–d. At pumping strength $D_{\text{max}} = D_1^{\text{th}} = 0.0124$ (red and blue filled circles in Fig. 2d), $\tilde{\omega}_0$ reaches the real-frequency axis, and $E_0(x) \sim \psi_0(x)$ turns on as the first lasing mode; the Petermann factor there is $K_0 \equiv |\langle \epsilon_c | \psi_0 \rangle|^2 / |\langle \epsilon_c | \psi_0^2 \rangle|^2 = 28$.

Response to Reviewer #2

In the manuscript, the authors introduce the periodic-inversion ab initio laser theory (PALT), which describes the behavior of multi-mode lasing near an exceptional point (EP), characterized by the coalescence of two eigenmodes. Same as in other non-Hermitian systems, the EP in a laser cavity emerges from the gain-loss tuning of the media. And this process is described by Eq. 4, as the population inversion density resides between the first and second lasing thresholds ($D_1 < D < D_2$).

This manuscript is well-motivated, self-consistent, and of high quality, making me pleased to recommend it for publication. However, before making a final decision, I have one question and one suggestion to offer.

Response: We thank the reviewer for the valuable feedback and the recommendation.

Suggestion:

I suggest adding a subfigure to Fig. 1, positioned before the existing two subfigures. The content for it can refer to Fig. S3c and the draft figure I have submitted. The purpose is to visually emphasize the key sentence in the first page:

"We identify two ingredients that promote comb formation: spectral degeneracy and the spatial coalescence of the mode profile."

With this subfigure, we can clearly depict spectral degeneracy (or close to it) through labeling small ω_d and suggest spatial coalescence through the intersecting eigen-spectra that characterize EP.

Response: We thank the reviewer for this suggestion. The spectral degeneracy is indeed the most important factor, but it is hard to accurately convey that in a figure because there is no sharp transition between an ordinary two-mode laser and an EP comb. Technically, an ordinary two-mode laser is also a limit cycle, just that the additional frequency components have a vanishingly small intensity because of the small ζ factor. Also, Fig. S3c only rigorously applies to the linear regime below the first lasing threshold D_1^{th} (where the lasing intensity is zero), yet the EP comb formation requires the laser to operate in the nonlinear regime (above D_1^{th} , with a non-zero lasing intensity before the comb can turn on).

To emphasize the importance of spectral degeneracy, we revised the sentence the reviewer mentioned (on page 1 of the paper) to the following to highlight the strong ω_d^{-4} dependence:

We find the efficiency of comb teeth generation, characterized by a ζ factor, to be enhanced by the spectral degeneracy with an inverse quartic dependence on the frequency difference and by the spatial coalescence of the mode profile.

Additionally, we added the following paragraph on page 4 to comment on the lack of a sharp transition:

Note there is no sharp transition between an ordinary two-mode laser and an EP comb. An ordinary laser operating in the two-mode regime away from degeneracies is a trivial limit cycle with two dominant frequency components and is also rigorously described by Eqs. (8)–(12). Such a laser features a small ζ factor, so the second threshold D_2^{th} from the stability analysis reduces to the SALT threshold (Supplementary Sec. 4), and the intensities of the additional frequency components ($m \neq 0, 1$) are small enough to be neglected. When ζ is raised, D_2^{th} smoothly moves, and the additional frequency components above D_2^{th} smoothly increase.

and revised the following sentence on page 1 to highlight the nonlinear regime requirement:

The dynamic inversion then nonlinearly couples different frequencies to produce a frequency comb above D_c^{th} (Fig. 1c).

Lastly, following the reviewer's suggestion and also the suggestion of Reviewer #1, we have moved Fig. S3 of the supplementary materials to the main text as a new Fig. 2 on page 5:

Fig. 2. Exceptional point in a gain-loss coupled cavity. **a**, A coupled 1D cavity separated by a distributed Bragg reflector (DBR), with gain in the left cavity and absorption in the right cavity. Gray-scale colors indicate the cold-cavity permittivity profile $\epsilon_c(x)$. Orange and blue shades show the gain and absorption profiles $D_p(x)$ and $\sigma(x)$, respectively. **b,c**, The two relevant eigenvalues, $\tilde{\omega}_0$ and $\tilde{\omega}_1$, of the linear operator $\hat{\mathcal{O}}(\omega)$ in Eq. (4) with a linear gain $D_0(x) = D_p(x)$, as a function of the pumping strength D_{max} and the length of the passive cavity, $L_2 = 1340 \text{ nm} + \Delta$. The two eigenvalues meet at an EP (green circle). The absorption in the passive cavity is $\sigma/\epsilon_0 = 4.9 \text{ ps}^{-1}$. The red and blue curves indicate $\tilde{\omega}_0$ and $\tilde{\omega}_1$ with $\Delta = 0$. **d**, Eigenvalue trajectories on the complex-frequency plane for $\Delta = 0$, with the orange arrows indicating the directions of increasing D_{max} . Open circles indicate $D_{\text{max}} = 0$, and filled red and blue circles indicate the first lasing threshold $D_{\text{max}} = D_1^{\text{th}}$ where $\tilde{\omega}_0$ reaches the real-frequency axis. Dashed lines show the would-be above-threshold trajectories in the absence of gain saturation.

EP is also recognized as a phase transition point, such as the transition between PT-exact/broken phase. Even without PT symmetry, the discontinuous eigen-spectra (Fig. S3c) signify a transition in mode profiles. Is the frequency comb influenced by this transition, or just by ω_d ?

Response: We thank the reviewer for raising this interesting question. After looking into it, we think the PT-exact/broken transition does not play a role here because it is suppressed by the nonlinear gain saturation. To discuss this in the single-mode regime (before the comb turns on), we added the following paragraph on page 5 of the main text:

Above the first threshold ($D_1^{\text{th}} < D_{\text{max}} < D_2^{\text{th}}$), the red and blue dashed lines in Fig. 2b–d show the would-be eigenvalue trajectories with a hypothetical linear gain, in which case the system enters a PT-broken phase where one mode is localized in the pumped cavity and the other mode is localized in the lossy cavity. Gain saturation, however, clamps the saturated gain at the same level as the overall loss, which fixes the two nonlinearity-frozen eigenvalues $\{\tilde{\omega}_0, \tilde{\omega}_1\}$ near where they are at D_1^{th} (red and blue filled circles in Fig. 2d), and this single-mode laser stays close to a nonlinear EP without entering the PT-broken phase.

In the comb regime ($D_{\text{max}} > D_2^{\text{th}}$), the situation can in principle be more complex, but we found the mode profiles to still remain approximately the same. To discuss this, we expanded the following sentence on page 5 of the main text:

The spatial profiles at different frequencies are almost identical (Fig. 3c); they remain comparable to the profiles near the EP in the single-mode regime $D_1^{\text{th}} < D_{\text{max}} < D_2^{\text{th}}$ where the two modes almost coalesce, without entering the PT-broken phase.

Response to Reviewer #3

In this revised manuscript, the authors have effectively addressed the majority of the issues I raised. However, before recommending this paper for publication, I still have two concerns based on the authors' response.

Response: We thank the reviewer for the thorough review and valuable feedback.

Firstly, with regard to the potential impact of the research, the authors assert that the EP laser can be utilized for sensing. While it is acknowledged that enhanced sensitivity has been demonstrated using microlasers at EPs (Refs. [15,16]), there appears to be some ambiguity regarding the specific sensing improvements facilitated by the current work.

Response: We thank the reviewer for this question. In the context of sensing, what the current work brings is not a specific improvement in performance but the recognition that there exists a different regime of laser behavior, which was not known but must be accounted for when designing an EP-enhanced laser sensor. To elaborate on this point, we added the following paragraph in the Discussion section (on page 8 of the main text):

As EP sensors are more sensitive closer to an EP,^{13–17} it may be desirable to operate such a sensor as close to an EP as possible. This work shows that when an EP laser is brought sufficiently close to an EP, it necessarily develops into a comb above a pump threshold. In such a comb regime, the optimal sensing scheme and the parametric dependence are nontrivial and can be the subject of a future study.

Secondly, the authors have clarified that the repetition rate of the EP comb can be adjusted by varying different parameters. Nevertheless, the underlying physics governing this dependence remains unclear. It raises questions as to whether the comb lines are not in resonance with the cavity modes. Further clarification on this aspect would be beneficial for a comprehensive understanding of the findings.

Response: We thank the reviewer for this important question. It is difficult to extract the underlying physics directly from the equations that govern the repetition rate ω_d , but we can explain the parameter dependence with an approximation. We added the following sentences in the Discussion section on page 8 of the main text:

The repetition rate ω_d of the EP comb is determined by the stability eigenvalue problem (Supplementary Sec. 4) at the threshold D_c^{th} and by solving the nonlinear Eqs. (11)–(12) self-consistently above D_c^{th} . While it is hard to extract insights from these complex equations, empirically we found the distance between the two linear SALT eigenvalues of Eq. (4) to

provide a crude approximation, $|\omega_d| \approx |\tilde{\omega}_1 - \tilde{\omega}_0|$, near D_c^{th} . Close to an EP, the two linear eigenvalues are sensitive to all parameters of the system, so ω_d can be tuned by changing the absorption, coupling strength, refractive index, *etc.*

We do know that the two center comb lines are in resonance with the two near-degenerate cavity modes, but the additional comb lines are not in resonance with additional cavity modes. To show this, we added the following figure to page 12 of the Supplementary Materials:

Supplementary Fig. 3. Comparison between the EP comb spectrum (upper panels) and the resonant frequencies of the active cavity from SALT through Eq. (S28) (lower panels). $\tilde{\omega}_0$ and $\tilde{\omega}_1$ form the near-EP pair. The next nearest resonant modes, $\tilde{\omega}_2$ and $\tilde{\omega}_3$, are separated from $\tilde{\omega}_{0,1}$ by the free spectral range of the active cavity. The right panels show zoom-ins near the EP frequency.

with the following accompanying paragraph in Supplementary Sec 7:

Supplementary Fig. 3 compares (1) the comb spectrum in Fig. 3g of the main text at $\Delta = 0$, $D_{\text{max}} = 0.2$, $\sigma/\varepsilon_0 = 8.1 \text{ ps}^{-1}$ with (2) the nonlinearity-frozen active-cavity eigenvalues $\{\tilde{\omega}_n\}$ of Eq. (S28) for the same parameters and using the lasing mode \mathbf{E}_0 from SALT (which predicts the system to stay single-mode at $D_{\text{max}} = 0.2$). We see that the two center comb lines $\{\omega_0, \omega_1\}$ lie close to the two near-degenerate active-cavity eigenvalues, $\{\text{Re}(\tilde{\omega}_0), \text{Re}(\tilde{\omega}_1)\}$. The other comb lines do not line up with any additional cavity modes (which are far away in frequency) since they are generated by the nonlinear gain through dynamic four-wave mixing rather than by the other cavity modes. Note that $\text{Im}(\tilde{\omega}_1)$ has moved down on the complex-frequency plane (*i.e.*, becomes more lossy) compared to its value at the comb threshold $D_{\text{max}} = 0.064$ (see Fig. 2d of the main text) because of the increased absorption σ .

We also added the following sentence on page 5 of the main text to mention it:

The two center comb lines $\{\omega_0, \omega_1\}$ lie close to the two near-degenerate active-cavity resonances from SALT in Eq. (4), $\{\text{Re}(\tilde{\omega}_0), \text{Re}(\tilde{\omega}_1)\}$; the remaining comb lines are generated by the nonlinear gain through four-wave mixing and are not lined up with any additional cavity modes (Supplementary Fig. 3).

REVIEWERS' COMMENTS

Reviewer #1 (Remarks to the Author):

The authors have significantly improved the manuscript, addressing all prior concerns. Given the thorough theoretical analysis and the relevant connection to the Fano-laser work published in Nature Photonics, I believe the paper is now suitable for publication. Therefore, I recommend its acceptance for publication.

Reviewer #2 (Remarks to the Author):

In the manuscript, the authors theoretically studied the frequency comb generated by a multi-mode laser operating near an exceptional point. To understand its behavior, they developed PALT, an ab initio numerical method to solve the Maxwell–Bloch equations, which successfully models the steady state of a realistic exceptional laser example (AlGaAs cavity). As a researcher in numerical methods, I find this work neat and solid, leaving little to be doubted.

Throughout several rounds of review, I raised a series of questions that helped clarify the applicability of this theory. This was done with the following considerations: exceptional lasers represent a successful interdisciplinary application of non-Hermitian physics and nonlinear photonics. There are surely more intriguing phenomena to explore, so the boundaries of the theory should be clearly defined.

In sum, I have no further questions and therefore highly recommend the publication of this work.

Reviewer #3 (Remarks to the Author):

In this revised submission, the authors have thoroughly addressed all of my previous comments. I have no further concerns and recommend this paper for publication.